# MG2FLOWNET: ACCELERATING HIGH-REWARD SAMPLE GENERATION VIA ENHANCED MCTS AND GREEDINESS CONTROL

## ABSTRACT

Generative Flow Networks (GFlowNets) have emerged as a powerful tool for generating diverse and high-reward structured objects by learning to sample from a distribution proportional to a given reward function. Unlike conventional reinforcement learning (RL) approaches that prioritize optimization of a single trajectory, GFlowNets seek to balance diversity and reward by modeling the entire trajectory distribution. This capability makes them especially suitable for domains such as molecular design and combinatorial optimization. However, existing GFlowNets sampling strategies tend to overexplore and struggle to consistently generate high-reward samples, particularly in large search spaces with sparse high-reward regions. Therefore, improving the probability of generating high-reward samples without sacrificing diversity remains a key challenge under this premise. In this work, we integrate an enhanced Monte Carlo Tree Search (MCTS) into the GFlowNets sampling process, using MCTS-based policy evaluation to guide the generation toward high-reward trajectories and Polynomial Upper Confidence Trees (PUCT) to balance exploration and exploitation adaptively, and we introduce a controllable mechanism to regulate the degree of greediness. Our method enhances exploitation without sacrificing diversity by dynamically balancing exploration and reward-driven guidance. The experimental results show that our method can not only accelerate the speed of discovering high-reward regions but also continuously generate high-reward samples, while preserving the diversity of the generative distribution. All implementations are available at https://anonymous.4open.science/r/MG2FlowNet-68B2/.

## 1 INTRODUCTION

Generative Flow Networks (GFlowNets) (Bengio et al., 2021; Jain et al., 2022; Gao et al., 2022; Bengio et al., 2023; Zhang et al., 2025) have recently emerged as a powerful tool for generating diverse high-quality candidates by learning to sample from a reward-proportional distribution. This property makes GFlowNets particularly attractive for a wide range of structured generation tasks. For example, Jain et al. (2022) integrates GFlowNets into an active learning pipeline for biological sequence design. In the domain of Bayesian structure learning, Deleu et al. (2022) and Nishikawa-Toomey et al. (2022) employ GFlowNets to model posterior distributions over discrete compositional structures such as Bayesian networks. Liu et al. (2023a) utilizes GFlowNets for sampling modular subnetworks, improving model generalization under distributional shifts.

Despite these advantages, vanilla GFlowNets often struggle to efficiently discover high reward samples in complex environments. While their inherent exploratory nature enhances diversity, it may lead to excessive coverage of low-reward regions, particularly during early training when the sampling policy lacks guidance and relies on self-collected experience. This results in slow convergence and suboptimal performance in sparse reward scenarios. The fundamental challenge lies in balancing broad exploration with efficient high-reward discovery, highlighting the need for directed exploration strategies that maintain diversity while effectively guiding the model toward high-reward areas. To address these issues, several recent efforts have incorporated reinforcement learning techniques into the GFlowNets framework. Notably, QGFN (Lau et al., 2024) introduces action value (*i.e.*, $Q$-value) to enhance backward policy estimation, while another approach applies MCTS and

maximum entropy regularization (Morozov et al., 2024) to enhance planning capabilities. However, these strategies often rely on noisy or inaccurate value estimates in the early training stages and may still fail to effectively guide the model toward high-reward regions. Moreover, they typically lack a fine-grained mechanism for dynamically adjusting the trade-off between exploration and exploitation throughout training. This raises a key question: *How can we enhance the model's ability to explore high-reward regions early in training, while adaptively using historical experience in later stages to maintain high-reward sampling?*

Monte Carlo Tree Search (MCTS) is a best-first search algorithm that has demonstrated strong performance in sequential decision-making tasks such as AlphaGo Zero (Silver et al., 2016; 2018). It offers an effective way to explore large search spaces. We build on this idea by proposing a framework that integrates Polynomial Upper Confidence Trees (PUCT)-guided MCTS (Coulom, 2006; Kocsis & Szepesvári, 2006) with a tunable $\alpha$-greedy sampling strategy. As shown in Figure 1, the framework directs GFlowNets toward promising regions of the state space: in explored areas, `MG2FlowNet` favors actions leading to high-reward states, while in unexplored areas it still allocates probability mass to encourage the discovery of potentially valuable states. The $\alpha$-greedy mechanism combines the $Q$-values estimated from MCTS rollouts with the forward policy of GFlowNets, allowing adaptive control over the degree of greediness. This design improves the efficiency of reaching high-reward samples while preserving the diversity of exploration. Our main **contributions** are as follows:

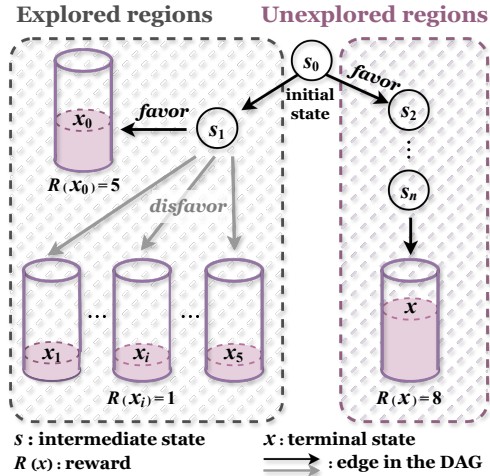

Figure 1: **Strategy of** `MG2FlowNet`. `MG2FlowNet` prioritizes high reward states in explored regions while still allocating effort to unexplored areas, ensuring that potential high reward states are not overlooked.

❶ We present a novel integration of enhanced **M**CTS and **G**reediness control with **GFlowNets** (termed `MG2FlowNet`), which demonstrates significant improvements in both sample efficiency and consistent generation of high reward samples, especially in large, sparse reward domains such as molecule design.

❷ We achieve a better balance between exploration and exploitation. By introducing the PUCT method in the selection phase of MCTS, we enable the model to adaptively adjust the intensity of exploration and exploitation.

❸ We implement a controllable soft greedy strategy. We consider both the $Q$-value of individual nodes and the flow distribution of the flow network, and use the distribution of $Q$-values as the greedy term, which enables us to achieve relatively good results even in the early training stage when $Q$-values are unstable.

❹ We empirically validate `MG2FlowNet` on several tasks, demonstrating improved sample efficiency and high-reward discovery while maintaining the diversity of generated solutions.

## 2 RELATED WORK

**GFlowNets.** Generative Flow Networks (GFlowNets) were first proposed by Bengio et al. (2021) as a framework for sampling compositional objects with probabilities proportional to their rewards, providing a scalable alternative to classical methods in multimodal or sparse reward settings. This formulation enables diverse and efficient exploration, which has proven useful in applications such as biological sequence design (Jain et al., 2022) and Bayesian structure learning (Deleu et al., 2022). Theoretical advances further connected GFlowNets to variational inference (Zimmermann et al., 2022). Despite these advances, classical GFlowNets are often prone to inefficient exploration, slowing convergence, and reducing the quality of high-reward samples. Our work addresses this drawback by introducing a mechanism that better balances exploration and exploitation.

**Improving GFlowNets Sampling.** MCTS has demonstrated strong performance in sequential decision making, most notably in AlphaGo and AlphaZero (Silver et al., 2016; 2018). A key refinement is the PUCT algorithm (Coulom, 2006; Kocsis & Szepesvári, 2006), which integrates visit counts

into the selection rule to balance exploration and exploitation. Inspired by these ideas, our work incorporates PUCT-guided MCTS and controllable greedy strategies into the GFlowNets framework, enabling more efficient trajectory generation while preserving theoretical guarantees.

## 3 PROBLEM FORMULATION

We generate a candidate object from the initial state $s_0$, making a sequence of actions to finally transfer the state to the terminal state $x$ with probability proportional to a reward function $R(x)$ : $x \to \mathbb{R}^+$. The state transformation process can be illustrated as a directed acyclic graph (DAG). We denote this sequence as a trajectory $\tau = (s_0 \to ... \to x)$, the set of complete trajectories as $\mathcal{T}$, the set of states as $\mathcal{S}$, the set of terminal states as $\mathcal{X}$, the action as $(s \to s')$, and the set of actions as $\mathcal{A} = \{(s \to s') | s, s' \in \mathcal{S}\}$. We say $s$ is a parent of $s'$, and $s'$ is a child of $s$. We denote the $C(s)$ as the set of children of $s$, the $P(s)$ as the set of parents of $s$. The set of available actions from $s$ is denoted as $\mathcal{A}(s)$, and thus $\mathcal{A}(x) = \varnothing$ for any terminal state $x$. For any state $s$, define the state flow $F(s) = \sum_{s \in \tau} F(\tau)$, and for any edge $s \to s'$, the edge flow $F(s \to s') = \sum_{\tau=(...\to s \to s' \to ...)} F(\tau)$. We denote the outflow of initial state $s_0$ as $Z = F(s_0) = \sum_{\tau \in \mathcal{T}} F(\tau)$. The forward and backward probabilities are denoted as $P_F$ and $P_B$:

$$P_F(s' \mid s) = F(s \to s')/F(s), \quad P_B(s \mid s') = F(s \to s')/F(s'). \tag{1}$$

We have the trajectory balance (TB) constraint (Malkin et al., 2022) for any complete trajectory $\tau = (s_0 \to ... \to s_n)$:

$$Z \prod_{t=1}^{n} P_F(s_t | s_{t-1}) = F(x) \prod_{t=1}^{n} P_B(s_{t-1} | s_t), \tag{2}$$

And the trajectory loss is defined as:

$$\mathcal{L}_{\text{TB}}(\tau) = \left( \log \frac{Z_\theta \prod_{t=1}^{n} P_F(s_t \mid s_{t-1}; \theta)}{R(x) \prod_{t=1}^{n} P_B(s_{t-1} \mid s_t; \theta)} \right)^2. \tag{3}$$

Our MCTS algorithm constructs an initially empty directed acyclic graph (DAG) and expands it incrementally. We denote the resulting MCTS-DAG by $\mathcal{G}_m = (\mathcal{V}_m, \mathcal{A}_m)$ and the GFlowNets sampling DAG by $\mathcal{G} = (\mathcal{V}, \mathcal{A})$, where $\mathcal{V}_m$ and $\mathcal{V}$ are the sets of nodes, and $\mathcal{A}_m \subseteq \mathcal{V}_m \times \mathcal{V}_m$, $\mathcal{A} \subseteq \mathcal{V} \times \mathcal{V}$ are the sets of directed edges (actions). In $\mathcal{G}_m$, individual nodes are denoted by $n$. To formalize the MCTS iteration process, we denote the expected value of taking action $a$ at node $n$ by $Q(n, a)$, and the number of times action $a$ has been executed at $n$ by $N(n, a)$. Let $T \subseteq \mathcal{V}_m$ be the set of terminal nodes, with each terminal node denoted by $n_T$, and let $F \subseteq \mathcal{V}_m$ be the set of fully expanded nodes (*i.e.*, nodes with children). Nodes without children are referred to as leaves. **The detailed notations are provided in Table 3.**

## 4 METHODOLOGY

**Framework.** To overcome the limitation that GFlowNets struggle to sample high-reward regions consistently, we incorporate modified MCTS for its planning capability and introduce a parameter $\alpha$ to link it with the flow network, thereby controlling the level of greediness. As illustrated in Figure 2, we start from the state $s_0$, which has several available actions $\{a_0, a_1, \ldots, a_n\}$. The objective is to choose the action most likely to guide the search toward high-reward regions. Before making this choice, we perform $I$ rounds of MCTS on $n_0$ (corresponding to the $s_0$ in $G$) in $G_m$. Each round consists of four phases: selection, expansion, simulation, and backpropagation. An iteration uses the current node history to identify a promising path, simulates it to a terminal state $n_T$, records the reward $R(n_T)$, and then backpropagates this reward to update the statistics $Q(n, a)$ and $N(n, a)$ of all nodes along the path. After $I$ iterations, actions from $s_0$ yield distinct $\{ Q(s_0, a_i) \mid a_i \in \mathcal{A}(s_0) \}$. We then apply a mixed strategy, controlled by $\alpha$, that combines $Q(n, a)$ with the prior $P_F$ to select an action and move from $s_0$ to $s_1$. This procedure repeats until a terminal state $x$ is reached, producing a trajectory $\tau = (s_0 \to s_1 \to \cdots \to x)$. We next detail the four stages and explain how our framework implements controllable greedy sampling.

### 4.1 PUCT GUIDED SELECTION

The selection phase aims to efficiently reach promising leaf nodes through a balance of exploration and exploitation. During the selection phase, we will meet these situations: (1) The current node has no child nodes and is called a leaf node. If it is a terminal node (corresponding to a terminal

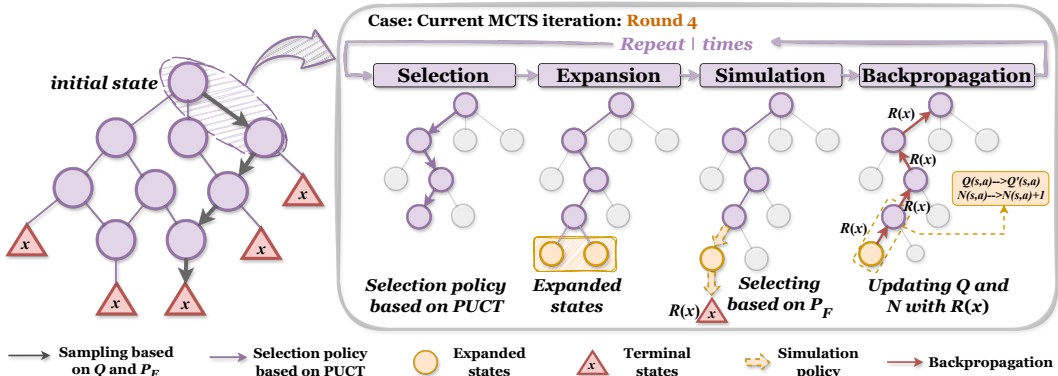

Figure 2: **Illustration of framework.** The left panel shows trajectory sampling in GFlowNets, where each action is chosen based on the updated $Q(s, a)$ and $P_F$ after $I$ rounds of MCTS iterations. The right panel illustrates the MCTS procedure, including selection, expansion, simulation, and backpropagation, with the fourth iteration shown as an example for clarity.

state $x \in \mathcal{X}$), then we return the reward $R(x)$ and start the backpropagation stage, and we use $-2$ in the code to indicate this case; else we add all legal child nodes to this node, which is the expansion phase, and we use $-1$ in the code to indicate this case. (2) The current node has child nodes. Define the $\pi(n, a)$ as the selection strategy for taking action $a$ from node $n$ to its child node $n'$. We define a selected trajectory $\tau = (n_0 \to ... \to n_i)$ from the node $n_0$ corresponding to the current state $s_0$ to the leaf node $n_i$.

$$\text{select}(n) = \begin{cases} (n, -2) & \text{if } n \in T \\ (n, -1) & \text{if } n \notin F \\ \text{select}(\pi(n, a)) & \text{if } n \in F \end{cases} \tag{4}$$

The selection strategy $\pi(n, a)$ during the selection phase warrants careful consideration. An overly greedy approach that relies solely on the $Q$-value predicted by $G_m$ may lead to local optima, while excessive exploration could slow down convergence. Therefore, we employ the PUCT formula to dynamically balance exploration and exploitation, using the exploration coefficient $c_{\text{puct}}$ to actively adjust exploration intensity. This enables rapid acquisition of high-quality samples while maintaining diversity, allowing for greedy generation of high-scoring samples without sacrificing variety.

$$\text{PUCT}(n, a) = Q(n, a) + c_{\text{puct}} \cdot P_F \cdot \frac{\sqrt{\sum_{a'} N(n, a')}}{1 + N(n, a)}, \tag{5}$$

$$\tilde{v}_a = \exp(\text{PUCT}(n, a) - \max_{a' \in A(s)} \text{PUCT}(n, a')), \tag{6}$$

$$p_a = \tilde{v}_a / (\sum_{a' \in A(s)} \tilde{v}_{a'}). \tag{7}$$

The visitation count $N(n, a)$ maintains the exploration statistics for action $a$ at state $n$. The exploration bonus term exhibits an inverse relationship with $N(n, a)$, creating an adaptive exploration-exploitation trade-off: When $N(n, a)$ is small (underexplored), the term dominates to encourage exploration. As $N(n, a)$ grows (well explored), the term decays to prioritize exploitation of high-reward actions. This dynamic balance motivates our selection policy $\pi(n, a) \sim \text{Categorical}(p_a)$, which follows a categorical distribution over the action space $\mathcal{A}(n)$.

## 4.2 EXPANDING ALL LEGAL ACTIONS

The expansion stage is a process of adding child nodes to the leaf node. Consider the following scenario: if only a single child node is expanded upon encountering a leaf node, each subsequent visit to this node would require another expansion operation until the node is fully expanded, resulting in significant computational overhead. To address this inefficiency, we propose expanding all legal child nodes during the expansion phase. This expansion approach is computationally justified because the exploration term in our PUCT formulation Eq. (5) guarantees that these newly created nodes will be properly prioritized based on their low $N(n, a)$, ensuring they will be systematically explored in future iterations. These expanded nodes' initial value of $Q(n, a)$ and $N(n, a)$ will be initialized to zero. Because these nodes lack historical statistics due to their initial state, the GFlowNets' forward probabilities $P_F$ naturally dominate the selection process in subsequent iterations. This design principle is explicitly illustrated in Eq. (5).

### 4.3 SIMULATION USING FORWARD PROBABILITY OF GFLOWNETS

To better leverage the model's forward transition probability $P_F$, we follow the forward transition probability in our simulation stage. Notably, in the selection stage, we already obtain a trajectory $\tau = (n_0 \to ... \to n_i)$, then we expand the children of $n_i$. After expansion, we select a node $n_e$ from these expanded children based on the $P_F$ inherent to GFlowNets. And then the simulation sampling process is performed starting from $n_e$ in line with $P_F$ until a terminal node $n_T \in T$ is reached. In the simulation stage, we similarly obtain a trajectory $\tau' = (n_e \to ... \to n_T)$ from the child node $n_e$ to the terminal node $n_T$. This trajectory $\tau'$ is designed to simulate which terminal nodes can be reached by node $n_e$, and during the subsequent backpropagation phase, the reward values from these terminal nodes are propagated backward along the selection trajectory $\tau$ to the root node. Through this process, we gain the capacity to anticipate and evaluate future states multiple steps ahead.

### 4.4 BACKPROPAGATION ALONG PROMISING PATHS

The $Q(n, a)$ update during backpropagation is crucial because it directly governs the accuracy of node evaluations in $G_m$. Our method employs weighted incremental updates for value backpropagation, where each node's $Q(n, a)$ and $N(n, a)$ are updated as:

$$Q(n, a) \leftarrow Q(n, a) + \frac{R(n_T) - Q(n, a)}{n_{visit}}, \quad n \in \tau(n_0 \to ... \to n_i), \tag{8}$$

$$n_{visit} \leftarrow n_{visit} + 1. \tag{9}$$

The update for $Q(n, a)$ and $N(n, a)$ is as shown above in Eq. (8) and Eq. (9). The update of $N(n, a)$ aims to control the level of exploration. As $N(n, a)$ increases, the exploration term in Eq. (5) decreases, leading to reduced exploration of that node. At the same time, the update of $Q(n, a)$ ensures that nodes with higher reward values are more likely to be selected. These two components are both essential and work together to strike a balance between exploration and exploitation. There is a challenge in updating the nodes during the backpropagation phase. Because updating different parents leads to a completely different distribution of the flow network. We address the backpropagation challenge by updating only the nodes along the trajectory $\tau$ selected during the selection phase. Details are provided in Sec. E.

### 4.5 GREEDINESS CONTROL

To combine the $P_F$ and the $Q(n, a)$ prediction accuracy for taking action $a$ at node $n$, we propose a $\alpha$-greedy strategy in Eq. (11) to dynamically adjust the proportion between the global flow network and the value distribution of $Q$-values.

$$p_i = \frac{(Q_i - Q_{min})}{\sum_k (Q_k - Q_{min})}, \tag{10}$$

As shown in Eq. (10), instead of adopting a max $Q$ strategy, we choose to use a softmax policy based on $Q$-values. This decision stems from our goal of encouraging more goal-directed behavior on top of a learned flow model, rather than pursuing greediness for its own sake. Directly using the maximal $Q$-value may lead to premature convergence and getting stuck in a local optimum. Moreover, $Q$-values are often inaccurate in the early training stages, making the model highly sensitive to estimation errors. To mitigate these risks, we opt for a soft $Q$-value policy that balances exploitation and exploration more effectively.

$$\mu \sim \text{Categorical}\left(\frac{(1 - \alpha) \cdot P_F + \alpha \cdot p}{\|(1 - \alpha) \cdot P_F + \alpha \cdot p\|_1}\right). \tag{11}$$

In summary, we can adaptively balance exploration and exploitation by PUCT during the selection phase. We can control the level of greediness in the model by tuning the value of $\alpha$, and consider both the global flow network and the value distribution, thereby making it adjustable and controllable. The details of our algorithm are as in Sec. C.

## 5 EXPERIMENTS

In this section, we evaluate the performance of `MG2FlowNet` on two tasks: **Hypergrid** and **Molecule Design**. These tasks are designed to test the model under different conditions: the former involves long action trajectories with sparse rewards, while the latter involves short trajectories with a large action space, also under sparse rewards. Our evaluation focuses on two central research questions:

❶ How effectively does the model achieve early discovery and sustained generation of high-reward candidates?

❷ How well does the model maintain diversity among generated candidates?

To provide a comprehensive view of the model's capabilities with respect to these questions, we report a set of carefully chosen metrics. In addition, we examine how key parameters of the model are learned, and we conduct ablation studies on the joint forward probability $P_F$ and MCTS-based planning in Sec. F. The experimental setup and results for each task are detailed below.

## 5.1 HYPERGRID TASK

**Task Description.** We begin our evaluation with the Hypergrid environment introduced by Bengio et al. (2021), a canonical testbed for assessing compositional generalization in GFlowNets. The environment consists of a $D$-dimensional discrete state space structured as a hypercube with edge length $H$, yielding $H^D$ distinct states. This task challenges agents to develop long-horizon planning capabilities while learning from extremely sparse reward signals. The agent initiates each episode at the origin $(0, 0, \cdots, 0) \in \mathbb{Z}^D$ and executes actions by incrementing any single coordinate by 1 (*i.e.*, $\Delta x_d = 1$ for dimension $d$). From any state, the agent may alternatively choose a termination action that yields a reward determined by the following function:

$$R(\mathbf{x}) = R_0 + R_1 \prod_{d=1}^{D} \mathbb{I}\left(\left|\frac{x_d}{H-1} - 0.5\right| \in (0.25, 0.5]\right) + R_2 \prod_{d=1}^{D} \mathbb{I}\left(\left|\frac{x_d}{H-1} - 0.5\right| \in (0.3, 0.4]\right), \quad (12)$$

where $\mathbb{I}$ denotes the indicator function, and we adopt the standard parameterization: $R_0 = 10^{-5}$, $R_1 = 0.5$, $R_2 = 2$, with grid parameters $H = 8$, $D = 4$.

**Metrics.** Since the grid environment is relatively simple with only 16 modes, we adopt the following two metrics to evaluate the performance of our model: 1) **Number of modes**, which reflects the model's exploration capacity and structural diversity. 2) **The $\ell_1$ error** $\mathbb{E}_{x \sim p}\left[\left|p(x) - \frac{R(x)}{Z}\right|\right]$, where $Z = \sum_x R(x)$, measuring how well the learned sampling distribution $p(x)$ matches the target reward distribution. This $\ell_1$ error directly assesses whether the GFlowNets achieves its fundamental objective of generating samples with probabilities proportional to their rewards.

**Baselines.** We compare `MG2FlowNet` with representative flow-based baselines like TB and MCMC (Malkin et al., 2022; Bengio et al., 2021; Zhang et al., 2022b), as well as several non-flow-based methods, including PPO (Schulman et al., 2017) and RANDOM-TRAJ (which samples actions uniformly at random). All methods are evaluated under the same grid environment and reward function to ensure fairness. The following sections present the results and analysis of `MG2FlowNet` in comparison to these baselines across multiple evaluation metrics.

**Effectiveness Evaluation.** The right of Figure 3 shows that the number of modes discovered is a key indicator of how effectively a model identifies high reward candidates. TB recovers 8 modes within 20,000 state visits, whereas `MG2FlowNet` achieves the same within only 10,000 visits. Most baselines eventually identify all 16 modes after 40,000 visits. These results indicate that `MG2FlowNet` is more effective at locating high reward regions. The underlying reason lies in the different exploration strategies: vanilla GFlowNets emphasize balanced exploration across all candidate regions, which slows down the process of reaching high-reward areas. In contrast, `MG2FlowNet` enhances the sampling process through action value prediction and controllable greedy exploration, which systematically biases the trajectories toward promising states. Once certain high-reward regions are discovered, the model tends to revisit and exploit those areas more frequently in subsequent iterations, thereby accelerating the discovery of near-optimal solutions and reducing the number of visits required to recover all modes.

**Accuracy Evaluation of GFlowNets.** The left of Figure 3 reports the $\ell_1$ error across different models, which captures the alignment between the generated distribution and the reward-proportional objective of GFlowNets. Vanilla GFlowNets achieve relatively low $\ell_1$ error by strictly adhering to the proportionality principle, ensuring that sampling frequencies closely follow reward magnitudes. By contrast, `MG2FlowNet` incorporates the MCTS algorithm to guide the sampling process, which introduces a more greedy bias toward high reward regions. As a consequence, once the model identifies promising areas, it tends to allocate a greater proportion of its sampling budget

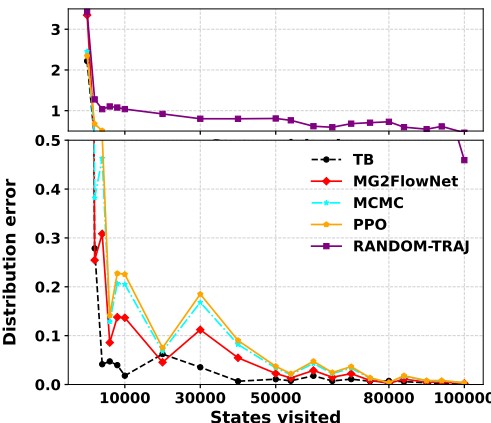 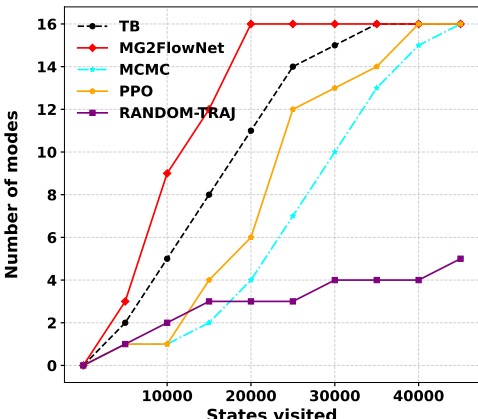

Figure 3: **High Reward Mode Discovery and Distribution Matching Error on Hypergrid.** *Left:* Comparison of the number of high-reward region modes that different models can find with the same number of visits. *Right:* Comparison of $\ell_1$ loss across models, measuring deviation between learned sampling distribution and target reward distribution.

to those regions in later training stages. This intentional departure from exact proportionality leads to slightly larger $\ell_1$ error values, but remains consistent with the design goal: prioritizing the rapid identification of promising regions and the generation of high reward candidates. In practice, this results in a sampling distribution that, while not perfectly reward proportional, is better suited for producing near-optimal solutions within fewer training rounds.

## 5.2 Molecule Design Task

**Task Description.** Recent advances in artificial intelligence have revolutionized computational chemistry, particularly in molecular property prediction and design (Du et al., 2024; Li et al., 2024; Zhang et al., 2023b). Molecular design presents an ideal application scenario for GFlowNets, as it requires simultaneous optimization of two critical objectives: 1) *quality* (achieving target chemical properties) and 2) *diversity* (generating structurally distinct candidates). This dual requirement stems from practical drug discovery needs, where viable candidates must not only exhibit strong binding affinities but also possess synthesizable structures. We focus on the specific challenge of designing molecules with maximal binding energy to a target protein. To this end, we formally describe the action space for molecular generation: building on junction tree-based molecular generation and following Bengio et al. (2021), we define:

$$\mathcal{A}(s) = \{(v, b) \mid v \in \mathcal{V}(s), b \in \mathcal{B}\}, \tag{13}$$

where $v$ denotes the choice of target atom, $b$ denotes the choice of building block. where $\mathcal{V}(s)$ denotes attachable atoms in state $s$ and $\mathcal{B}$ is our building block vocabulary ($|\mathcal{B}| = 105$). Given a molecule, a building block can be added to the molecule at different positions. The combinatorial action space poses significant exploration challenges while enabling the generation of diverse molecular scaffolds.

**Metrics.** 1) **Number of modes**, which reflects the model's exploration capacity and structural diversity. 2) **Average top 100**, Among all generated candidate molecules, we report how many molecular states are visited when the top-100 average reward exceeds 7.0, 7.5, and 8.0. Fewer visited states to reach the corresponding average reward indicate faster discovery of high reward regions. 3) **Tanimoto similarity**, in molecular generation tasks, if all high-reward molecules produced are structurally almost identical, then even high reward values would indicate that the model suffers from mode collapse. To address this, we additionally adopt the Tanimoto similarity metric, which measures the structural differences among generated molecules and further reflects whether the model can maintain diversity while consistently generating high-reward molecules.

**Baselines.** we compare `MG2FlowNet` with four popular flow-based baselines, TB (Malkin et al., 2022), SubTB (Madan et al., 2023), DB (Bengio et al., 2021) and QGFN (Lau et al., 2024). Here, we adopt the same training parameters as the vanilla GFlowNets.

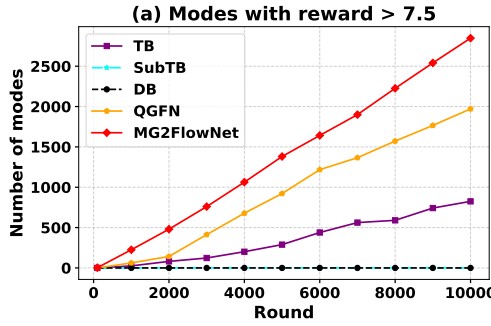 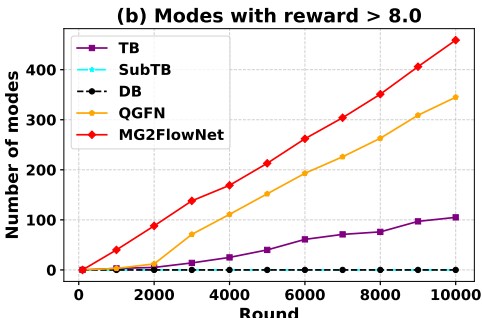

Figure 4: **Number of modes with reward** $> 7.5$ **and** $> 8.0$ **in molecule design task.** *Left*: Comparison of different models in terms of the number of modes with reward greater than 7.5. *Right*: Comparison of different models in terms of the number of modes with reward greater than 8.0.

**Effectiveness Evaluation.** Figure 4 reports the discovery of high reward samples. SubTB and DB show significantly inferior performance, as no high reward samples (with reward $> 7.5$ or $> 8.0$) are discovered even after 10,000 iterations. TB performs slightly better than SubTB and DB, but it still lags behind `MG2FlowNet` and QGFN in efficiently identifying high-reward samples. Notably, `MG2FlowNet` demonstrates superior effectiveness by discovering high-reward samples earlier and more consistently. In particular, `MG2FlowNet` surpasses QGFN in locating samples with reward $> 8.0$, successfully achieving this within only 300 iterations. This improvement stems from the exploration term in our Eq. (5) formulation, which plays a crucial role in the early training phase by adaptively balancing exploration and exploitation, unlike QGFN, which solely relies on $Q$-values.

Table 1 further confirms these observations with the average top 100 (avg top 100) rewards. SubTB and DB are excluded, as reaching average top 100 rewards of $7.0$, $7.5$, or $8.0$ would require prohibitively many state visits. Both QGFN and `MG2FlowNet` show marked improvements, benefiting from action value guided sampling. However, due to inaccurate $Q$-values in the early stages, QGFN some-

Table 1: Number of states visited for top candidates (lower is better). Bold: best; underline: second best.

| States visited | avg top 100 $> 7.0$ | avg top 100 $> 7.5$ | avg top 100 $> 8.0$ |
|---|---|---|---|
| TB | 2,824 | 6,425 | 12,816 |
| QGFN | 2,000 | 2,800 | 10,800 |
| MG2FlowNet | **644** | **964** | **5,204** |

times overestimates intermediate reward regions and tends to waste effort on low-potential paths. In contrast, `MG2FlowNet` performs consistently well across all thresholds ($7.0$, $7.5$, and $8.0$), strongly supporting the effectiveness of integrating MCTS and the $\alpha$-greedy strategy into the GFlowNets framework. These mechanisms enable adaptive balancing throughout training, ultimately leading to faster and more stable discovery of high-reward samples.

**Diversity Evaluation.** Diversity is assessed through the Tanimoto similarity in Figure 5, which reflects the similarity among generated samples. The results show that MG2FlowNet maintains a low Tanimoto similarity, indicating that even while adopting a greedy sampling strategy, it still preserves diversity to a large extent. This complements the evidence from the number of modes: although our model discovers substantially more modes than baselines, these are not redundant or trivial repetitions, as the Tanimoto similarity does not increase significantly. Together, these findings demonstrate that MG2FlowNet not only generates more high-reward candidates but also achieves this while maintaining diverse coverage of the search space.

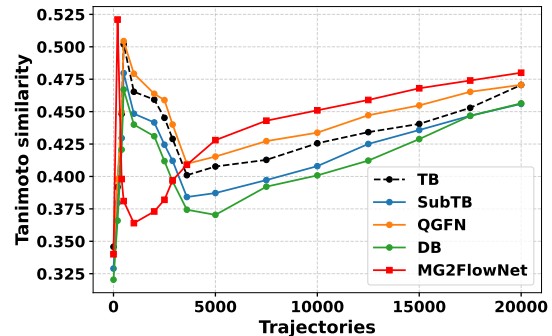

Figure 5: The Tanimoto similarity among the top-1000 molecules with the highest rewards generated by different models.

## 5.3 Ablation Study of Greediness Coefficient $\alpha$

We conduct ablation studies to investigate the role of the MCTS component in our framework, and to analyze how controlling the degree of integration between MCTS and GFlowNets influences model performance. As illustrated in the Table. 2, We investigate the effect of different greediness coefficients on model performance. Additionally, we introduce a temperature coefficient to realize a dynamic strategy that promotes stronger exploration during the early phase of training and gradually shifts toward greedier exploitation in later stages.

Table 2: Number of modes (average top $> 8.0$) discovered under different $\alpha$ settings. **Bold:** best in each column. *Temp* denotes that $\alpha$ is linearly annealed from 0 to 0.2 during training.

| Model settings | 12,000 | 24,000 | 40,000 |
|---|---|---|---|
| | # of modes (avg top $100 > 8.0$) | | |
| $c\_puct = 1, \alpha = 0.2$ | **92** | **175** | **459** |
| $c\_puct = 1, \alpha = 0.4$ | 17 | 46 | 249 |
| $c\_puct = 1, Temp$ | 1 | 26 | 103 |
| $c\_puct = 1, \alpha = 0$ | 14 | 61 | 105 |

**Greedy sampling ($\alpha = 0.4$).** We initially hypothesized that a higher value of the $\alpha$-greedy parameter, which corresponds to a more exploitative sampling strategy, would lead to better performance in the early stage of training but worse performance later on. The intuition was that a greedier policy would favor seemingly high-reward samples at the beginning, thus boosting the average top-$k$ score temporarily. However, the empirical results contradicted our expectations. We attribute this to excessive greediness, which causes premature convergence to suboptimal regions that initially appear promising but in fact correspond to low-reward trajectories, ultimately degrading long-term performance. This observation confirms that setting $\alpha$ too high has a detrimental impact on performance.

**Temperature controlled strategy ($\alpha$ from $0$ to $0.2$).** The temperature-controlled strategy theoretically offers more robust training dynamics. However, the empirical results diverged substantially from our expectations. We consider that this discrepancy arises from the configuration of the transition steps. Since determining the optimal value for this setting is nontrivial and beyond the primary scope of this work, we did not further pursue this direction. Importantly, this does not affect the core performance of our model, as the analysis was conducted as an auxiliary study. We attribute the observed performance degradation to the initially very small value of $\alpha$, which effectively reduces the model to a standard GFlowNets. Because our PUCT-based selection still incorporates an exploration term that dominates in the early stages of training, as discussed in our analysis of $c_{\text{puct}}$, the result is additional over-exploration on top of the base GFlowNets behavior, leading to a significant performance drop. From these observations, we conclude that the temperature-based scheduling of $\alpha$ is undesirable for two reasons: first, it is difficult to precisely control the rate of change; and second, the model often exhibits negative performance gains in the early phase. Consequently, we empirically determine that a fixed value of $\alpha = 0.2$ provides the best trade-off.

**Effect of MCTS.** When $\alpha = 0$, the model degenerates to a vanilla GFlowNets, yielding fewer modes than $\alpha = 0.2$ or $\alpha = 0.4$, which confirms the utility of the MCTS component in guiding sampling toward high reward regions. Although scheduling $\alpha$ to increase over training rounds leads to worse performance than $\alpha = 0$, this can be explained by excessive exploration on an unstable flow network in the early stage, causing a significant drop in performance. These results validate the effectiveness of the MCTS component.

## 6 Conclusion

In this paper, we introduce `MG2FlowNet`, a novel framework that integrates enhanced MCTS with controllable greediness into GFlowNets by adapting the selection, simulation, and backpropagation stages to DAG-structured environments. Our method employs a PUCT-based selection policy together with a tunable greediness mechanism to achieve a principled balance between exploration and exploitation. Through extensive experiments, we demonstrate that `MG2FlowNet` substantially improves both sample efficiency and diversity, particularly in large-scale and sparse-reward settings such as molecular generation. Overall, our study highlights the feasibility and effectiveness of combining MCTS with GFlowNets, providing insights for developing more powerful reinforcement learning algorithms integrated with GFlowNets. We also envision extending this approach or other more effective methods to dynamic environments where the action space and reward function evolve over time, thereby addressing more challenging tasks.

## ETHICS STATEMENT

This research relies exclusively on publicly available benchmark environments from Malkin et al. (2022), including the Hypergrid task and standard molecular design datasets, which contain no personally identifiable or sensitive information. No human or animal subjects were involved, and therefore no ethical approval was required. We acknowledge that generative modeling techniques, such as MG2FlowNet, could be misapplied in high-stakes domains, including drug discovery or personalized recommendation systems. However, our contributions are purely methodological, and all experiments are restricted to controlled and widely accepted benchmarks. To mitigate risks, we emphasize that any downstream applications of this method should be accompanied by domain-specific safeguards, rigorous evaluation, and appropriate human oversight. No conflicts of interest or external influences are associated with this work.

## REPRODUCIBILITY STATEMENT

We have taken extensive measures to ensure the reproducibility of our results. The main text and appendix provide full details of model architectures, optimization objectives, training hyper-parameters, and evaluation metrics. Additional experimental settings, ablation studies, and environment specifications are documented in the supplementary material. We have released the anonymized source code, configuration files, and preprocessing scripts, which are available at https://anonymous.4open.science/r/MG2FlowNet-68B2/. With the released resources and instructions, independent researchers are able to reproduce all reported results reliably.

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

CONTENTS

## A NOTATION

This section provides a summary of the key notations throughout this paper. The symbols and corresponding descriptions are listed in Table 3.

Table 3: Summary of Key Notations

| Symbol | Description |
|---|---|
| $s_0, s', s, s_t$ | States (initial state $s_0$, intermediate state $s', s, s_t$) |
| $x$ | Terminal state |
| $\mathcal{S}$ | State space |
| $\mathcal{X}$ | Set of terminal states |
| $\mathcal{A}(s)$ | Available action set at state $s$ |
| $\tau = (s_0 \to \cdots \to x)$ | A trajectory from the initial state $s_0$ to a terminal state $x$ |
| $\mathcal{T}$ | Set of trajectories |
| $F(s)$ | Flow of state $s$ (inflow equals outflow) |
| $F(s \to s')$ | Flow from state $s$ to $s'$ |
| $Z$ | Flow of the initial state $s_0$ |
| $P_F(s'|s)$ | Forward transition probability from $s$ to $s'$ |
| $P_B(s|s')$ | Backward transition probability from $s'$ to $s$ |
| $\mathcal{L}_{\text{TB}}(\tau)$ | Trajectory Balance loss |
| $n_T$ | Terminal node in $\mathcal{G}_m$ |
| $n, n_0$ | nodes in $\mathcal{G}_m$ (intermediate node $n$, root node $n_0$) |
| $Q(n, a)$ | Estimated value of taking action $a$ at node $n$ in $\mathcal{G}_m$ |
| $N(n, a)$ | Visit count of taking action $a$ at node $n$ in $\mathcal{G}_m$ |
| $R(x)$ or $R(n_T)$ | Reward of terminal state or node |
| $\pi(n, a)$ | Policy for selecting action $a$ at node $n$ |
| $\text{PUCT}(n, a)$ | PUCT value of taking action $a$ at node $n$ in $\mathcal{G}_m$ |
| $c_{\text{puct}}$ | Exploration coefficient in PUCT |
| $\alpha$ | Greediness coefficient |
| $\tilde{v}_a$ | Unnormalized score of action $a$ before softmax normalization |
| $p_a$ | Probability of selecting action $a$ after softmax normalization |
| $\ell_1$ | $\ell_1$ error between generated distribution and reward-proportional distribution |
| $\mathcal{V}(s)$ | Set of attachable atoms in state $s$ (molecule design) |
| $\mathcal{B}$ | Building block vocabulary for molecule generation |
| $\mathcal{G}$ | DAG representing the flow network |
| $\mathcal{G}_m$ | DAG representing the MCTS policy |

## B BACKGROUND

### B.1 GENERATIVE FLOW NETWORKS (GFLOWNETS)

GFlowNets (Bengio et al., 2021) are a class of generative models designed for sampling compositional objects $x \in \mathcal{X}$ through a sequential construction process. The generation process is formalized as a trajectory $\tau = (s_0, \ldots, x)$ over a directed acyclic graph (DAG) $\mathcal{G} = (\mathcal{S}, \mathcal{A})$, where $\mathcal{S}$ represents

the set of partially constructed states and $\mathcal{A} \subset \mathcal{S} \times \mathcal{S}$ denotes valid transitions (*e.g.*, adding a fragment to a molecule). The DAG is rooted at a unique initial state $s_0$, and terminal states correspond to fully constructed objects.GFlowNets are trained to satisfy flow balance conditions, ensuring that the flow $F(s)$ through states is conserved. Terminal states act as sinks, absorbing flow $R(s)$ (a non-negative reward), while intermediate states balance incoming and outgoing flows. This is expressed by the balance equation for any partial trajectory $(s_n, \ldots, s_m)$:

$$F(s_n) \prod_{i=n}^{m-1} P_F(s_{i+1}|s_i) = F(s_m) \prod_{i=n}^{m-1} P_B(s_i|s_{i+1}), \tag{14}$$

where $P_F$ and $P_B$ are the forward and backward policies, respectively, representing the fraction of flow directed toward children or parents of a state. For terminal states, $F(s) = R(s)$. $P_F$ and $P_B$ are related to the Markovian flow $F$ as follows:

$$P_F(s' \mid s) = \frac{F(s \to s')}{F(s)}, \quad P_B(s \mid s') = \frac{F(s \to s')}{F(s')} \tag{15}$$

### B.2 Monte Carlo Tree Search

Monte Carlo Tree Search (MCTS) (Coulom, 2006) is a best-first search algorithm that combines tree search with Monte Carlo simulation. The algorithm iteratively builds a search tree through four key phases: Selection → Expansion → Simulation → Backpropagation.

- **Selection**: Traverse the tree from root to leaf using a tree policy (typically Upper Confidence Bound for Trees, UCT) (Kocsis & Szepesvári, 2006):

$$a^* = \underset{a}{\mathrm{argmax}} \left( Q(s, a) + c\sqrt{\frac{\ln N(s)}{N(s, a)}} \right), \tag{16}$$

  where $Q(s, a)$ is the action value, $N(s)$ and $N(s, a)$ are visit counts, and $c$ is an exploration constant.
- **Expansion**: When reaching an expandable node, create one or more child nodes representing possible state transitions.
- **Simulation**: Perform a Monte Carlo rollout from the expanded node using a default policy to estimate the reward.
- **Backpropagation**: Update statistics along the traversed path:

## C Detailed Algorithm

This section describes the detailed algorithmic flow of our framework, as shown in the Algorithm 15.

## D Related Work

### D.1 Generative Flow Networks (GFlowNets)

Since their introduction by Bengio et al. (2021), GFlowNets have attracted increasing attention as a framework for sampling compositional objects with probabilities proportional to their rewards. This formulation enables efficient exploration in multimodal or sparse reward settings, where traditional approaches often struggle. Subsequent research has expanded both the theoretical foundations and methodological scope of GFlowNets. For instance, Malkin et al. (2022) and Zimmermann et al. (2022) connected GFlowNets to variational inference, showing advantages when leveraging off-policy data. Methodological improvements have focused on more efficient credit assignment (Pan et al., 2022; 2023b), while others explored multi-objective generation (Jain et al., 2022) and world modeling (Pan et al., 2023c). Extensions to unsupervised learning (Pan et al., 2023a) and bias reduction via isomorphism tests (Ma et al., 2024) have further broadened their applicability. From a probabilistic modeling perspective, Zhang et al. (2022b) proposed joint training of energy-based models

---

**Algorithm 1** MCTS Iterations with Greediness Controlled Sampling

---

1: **Input:** Reward function $R : \mathcal{X} \to \mathbb{R}_{>0}$, batch size $M$, model $P_F$ with parameters $\theta$, root node $n_0$, number of MCTS iterations $n_{\text{playout}}$, PUCT exploration coefficient $c_{\text{puct}}$, greediness factor $\alpha$
2: **Output:** MCTS sampling policy
3: Initialize MCTS graph $\mathcal{G}_m$ with root node $n_0$ corresponding to state $s_0$
4: **for** $i = 1$ to $n_{\text{playout}}$ **do**
5:     **Selection:** Traverse tree from $n_0$ to a leaf $n_i$ using PUCT; record path $\tau = (n_0 \to \cdots \to n_i)$
6:     **if** $n_i$ is terminal **then**
7:         **Backpropagate** reward $R(n_i)$ along the trajectory $\tau$
8:     **else**
9:         **Expansion:** Add $\mathcal{A}(n_i)$ (all available actions of $n_i$) to $\mathcal{G}_m$
10:        **Simulation:** Choose one child $n_e$ from the children generated during the expansion stage, and roll out to terminal node $n_T$ using $P_F$
11:        **Backpropagation:** Propagate reward $R(n_T)$ along $\tau$
12:     **end if**
13: **end for**
14: **Sampling Phase:** Use $\alpha$-greedy over $Q$-values predicted by $\mathcal{G}_m$ and $P_F$ to generate new samples
15: $\mu \sim \text{Categorical}\left( \frac{(1-\alpha) \cdot P_F + \alpha \cdot p}{\|(1-\alpha) \cdot P_F + \alpha \cdot p\|_1} \right)$

---

and GFlowNets, and subsequent work connected GFlowNets with diffusion models (Zhang et al., 2022a; Lahlou et al., 2023; Zhang et al., 2023a). Despite these advances, a persistent limitation of classical GFlowNets is their tendency toward inefficient exploration, which slows convergence and reduces the quality of high-reward samples. This work directly targets this drawback by proposing a principled mechanism to improve exploration efficiency.

### D.2 REINFORCEMENT LEARNING AND MCTS IN GFLOWNETS

Beyond standalone developments, recent efforts have explored integrating reinforcement learning techniques into GFlowNets, such as QGFN (Lau et al., 2024) and MaxEnt RL connections (Morozov et al., 2024). Relatedly, Monte Carlo Tree Search (MCTS) has achieved remarkable success in sequential decision-making, as demonstrated in AlphaGo and AlphaZero (Silver et al., 2016; 2018). A central refinement of MCTS is the Polynomial Upper Confidence Trees (PUCT) algorithm (Coulom, 2006; Kocsis & Szepesvári, 2006), which balances exploration and exploitation by incorporating visit counts. However, existing strategies, such as the $p$-greedy rule:

$$\pi_{\text{tree}} (\cdot \mid s) = (1 - p_s) \, \text{Softmax} \left( Q_{\text{tree}} (s, \cdot) \right) + p_s \cdot \mathcal{U}(C(s)), \tag{17}$$

which often fall into local optima when high-scoring nodes dominate $Q$ values, and the uniform exploration term ignores prior probabilities from GFlowNets. Entropy regularization has been proposed as a remedy, but it passively enforces exploration without leveraging historical statistics such as visit counts. Inspired by these limitations, we incorporate PUCT-guided selection and controllable greedy strategies into the GFlowNets framework, enabling more efficient trajectory generation while preserving theoretical guarantees.

### D.3 POTENTIAL APPLICATIONS OF GFLOWNETS

GFlowNets have demonstrated significant potential across multiple domains due to their unique ability to sample diverse solutions while maintaining reward proportionality. Their strong generalization capabilities enable effective handling of unseen states, making them particularly suitable for exploration-intensive tasks. The technology has shown remarkable success in molecular design, where it outperforms traditional reinforcement learning methods in exploring chemical space while preserving synthetic feasibility and drug-like properties.

Additionally, GFlowNets have emerged as a powerful framework for recommendation systems, demonstrating particular effectiveness in addressing the critical diversity-quality trade-off. Recent studies have successfully applied GFlowNets to enhance listwise recommendations by maintaining recommendation quality while significantly improving diversity Liu et al. (2023b), as well as optimizing user retention through intelligent exploration strategies Liu et al. (2024). Beyond traditional recommendation tasks, GFlowNets have shown remarkable adaptability for large language model

fine-tuning across various domains. Notable applications include diverse text generation for sentence infilling and chain-of-thought reasoning Hu et al. (2023), adversarial prompt generation in red teaming scenarios Lee et al. (2024), and complex puzzle-solving in domains such as BlocksWorld and Game24 Yu et al. (2024). These applications collectively demonstrate GFlowNets' versatility in handling both recommendation tasks and language model optimization challenges.

## E    BACKPROPAGATION CHALLENGE DETAILS

There is a challenge in updating the nodes during the backpropagation phase. Because updating different parents leads to a completely different distribution of the flow network. There are some alternative options: 1) The reward $R(n_T)$ is uniformly propagated back to each parent node, such that if there are $n$ parent nodes, each parent node updates its own $Q$-value with $R(n_T)/n$. 2) Distribute the reward $R(n_T)$ to all parent nodes proportionally based on their relative flow magnitudes within the flow network. Each parent node updates its own $Q$-value with $\rho R(n_T)$, where $\rho$ represents the proportion of flow from the parent node to a specific child node. 3) the reward $R(n_T)$ is only propagated back along the trajectory $\tau = (n_0 \to ... \to n_i)$ in the selection phase, described in detail in Sec. 4.1. Each node included in the trajectory $\tau$ updates its own $Q$-value with $R(n_T)$.

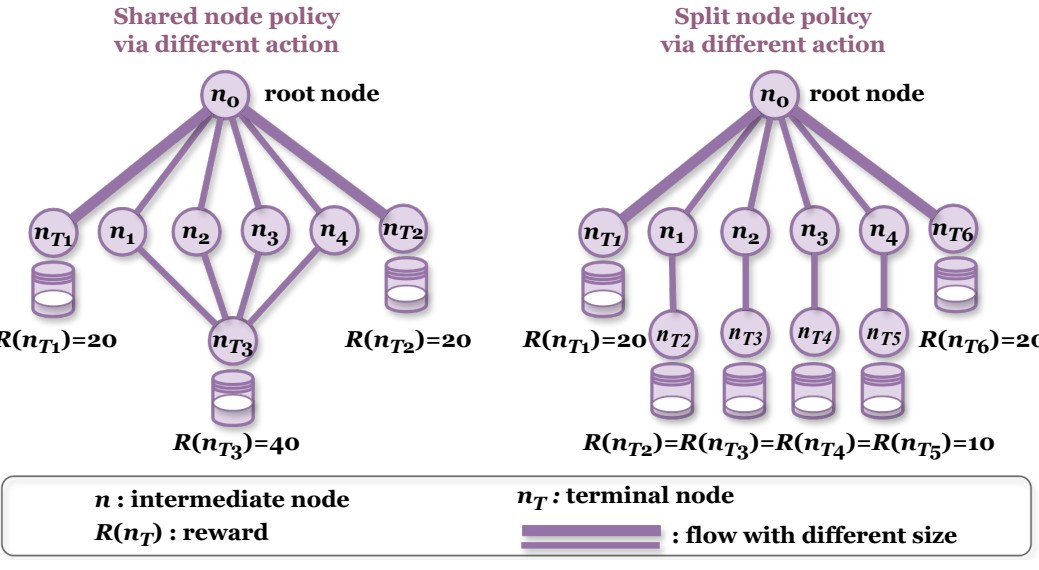

Figure 6: **Comparison of different representations for reaching the same state via multiple action sequences**. On the *left*, identical states are represented by a single shared node; On the *right*, the same state reached through different action sequences is represented by distinct nodes.

In our work, we adopt the third method for the following reasons. The first two approaches require updating all parent nodes and iteratively propagating these updates further up the tree by updating the parents of parents and so on. This results in significant computational overhead. Moreover, the second method incurs additional cost by computing the proportion of flow from each parent to its children, which further increases the computational burden. Furthermore, restricting updates to nodes along the selected path $\tau$ serves to emphasize the most promising trajectory. In contrast, the first two methods would dilute the relative contribution of this most promising path by distributing credit more broadly, which is undesirable. For different actions that lead to the same node, we have designed a global mapping of state nodes. For identical states, only one node is preserved. This approach also aligns with the objective of flow network training. As shown in Figure 6, if we create multiple nodes for the same terminal state via different action orders, we will distribute the proportion of high reward regions among these nodes, which may prevent the MCTS tree from accurately reflecting the high reward characteristics of these terminal nodes. Given these considerations, we opt for the third approach in our experimental design.

## F ADDITIONAL EXPERIMENTAL RESULTS

### F.1 STUDY ON GREEDINESS TERM

In this section, we use the Molecule Design experiment as an example to illustrate the significant role of the greediness term. The performance is evaluated based on the average reward of the top 100 samples, comparing greedy and non-greedy variants of the policy. As reported in Table 4, we can observe that achieving the same average top-100 reward requires visiting a significantly larger number of states, indicating that the proportion of high-reward samples obtained during sampling is relatively low. This further sub-

Table 4: The number of states visited for top-performing candidates. Results with and without the exploration term.

| States visited | Average Top-100 | | |
|---|---|---|---|
| | $>7$ | $>7.5$ | $>8$ |
| MG2FlowNet (without $Q$) | 1524 | 2404 | 9204 |
| MG2FlowNet (with $Q$) | **644** | **964** | **5204** |

stantiates the critical importance of the greedy term in the algorithm. The greedy term plays a pivotal role in guiding the model to sample from high-reward regions of the state space. Without this greedy component, the model fails to consistently generate high-reward samples, which fundamentally contradicts the original design intent of our approach.

### F.2 STUDY ON EXPLORATION TERM

To illustrate the critical role of the exploration term, we use the Hypergrid experiment as an example. As shown in the ablation study results in Table 5, the model performs significantly worse when the exploration term is removed, with performance reduced to less than half of that achieved with the term included. This clearly demonstrates the importance of the exploration component.

Table 5: The number of states visited for different modes discovered. Comparison of results with and without the exploration term.

| States visited | 4 | 8 | 16 |
|---|---|---|---|
| MG2FlowNet ($c_{\mathrm{puct}} = 0$) | 6,416 | 19,216 | 49,616 |
| MG2FlowNet ($c_{\mathrm{puct}} = 0.2$) | **4,816** | **9,616** | **20,816** |

We conclude that the exploration term plays a crucial role, especially in the early stages of training. During this phase, the $Q$-values are still inaccurate and highly uncertain. Without the exploration term, the agent tends to exploit unreliable estimates, leading to unstable learning dynamics and ultimately degraded performance.

Table 6: **Number of modes discovered under two thresholds.** Bold numbers are the highest in their respective column.

| Configuration | average top $> 7.5$ | | | average top $> 8.0$ | | |
|---|---|---|---|---|---|---|
| | 12,000 | 24,000 | 40,000 | 12,000 | 24,000 | 40,000 |
| MG2FlowNet ($c_{\mathrm{puct}} = 0.5, \alpha = 0.2$) | 402 | 947 | 1,879 | 128 | 312 | 704 |
| MG2FlowNet ($c_{\mathrm{puct}} = 2, \alpha = 0.2$) | **546** | 923 | 1,645 | **189** | 301 | 612 |
| MG2FlowNet ($c_{\mathrm{puct}} = 1, \alpha = 0.2$) | 507 | **1,080** | **2,848** | 165 | **384** | **1,053** |
| MG2FlowNet ($c_{\mathrm{puct}} = 1, \alpha = 0.4$) | 74 | 365 | 2,106 | 21 | 102 | 783 |
| MG2FlowNet ($c_{\mathrm{puct}} = 1$, Temp) | 4 | 170 | 1,123 | 1 | 43 | 398 |

### F.3 DIFFERENT EXPLORATION COEFFICIENT $c_{puct}$

As reported in Table 6, empirical results reveal that reducing the exploration coefficient $c_{\mathrm{puct}}$ significantly impairs the model's ability to consistently generate high reward samples. This observation supports our initial hypothesis: a smaller $c_{\mathrm{puct}}$ limits the model's capacity to explore less-visited regions of the state space, potentially causing it to overlook promising high-reward areas during

the early training phase. On the other hand, setting $c_{\text{puct}}$ to a relatively large value leads to stronger early-stage performance, as it encourages broader exploration and facilitates early discovery of high-reward trajectories. However, as training progresses, such high exploration settings begin to exhibit diminishing returns and even hinder further progress. While early identification of promising regions might be expected to guide subsequent sampling toward them, the problem actually arises from an imbalance between exploration and exploitation. When $c_{\text{puct}}$ is set excessively large, the exploration term in the PUCT (Eq. (5)) selection formula dominates, leading to over-exploration and suboptimal convergence. Based on extensive empirical evaluations, we find that $c_{\text{puct}} = 1$ provides a favorable trade-off between exploration and exploitation throughout the training process.

### F.4 COMPARISON OF DIFFERENT EXPANDING STRATEGIES

In the main text, we mentioned that our expanding strategy is adding all child nodes to the unexpanded node, because the exploration term in our PUCT formulation (Eq. (5)) guarantees that these newly created nodes will be properly prioritized based on their low $n_{visit}$, ensuring they will be systematically explored in future iterations.

To better validate the rationality of our design, we conducted a comparative experiment in the Hypergrid environment, comparing the approach of expanding all child nodes versus expanding only one child node. The goal was to measure the number of states visited required to discover the same number of modes. If discovering the same modes requires visiting significantly more states, it indicates wasted MCTS iterations and lower state-visit efficiency, making such an approach less desirable. The results of this comparative experiment are shown below:

Since the strategy of expanding only one child node requires an impractically large number of state visits to discover all 16 modes (rendering it meaningless for comparison), we omit this result here. However, the state visit counts required for discovering 4 and 8 modes clearly demonstrate the infeasibility of single child expansion. Our results show that this approach significantly reduces exploration efficiency.

We attribute this inefficiency to the fundamental limitation of single child expansion: Each training iteration predominantly revisits previously explored nodes due to constrained graph width in MCTS. This severe restriction on new node access dramatically reduces the exploration space. Even in our grid experiment, the state visit counts reached alarming magnitudes, let alone in molecular experiments with exponentially larger state spaces where such costs would become computationally prohibitive.

Table 7: **The number of states visited for different modes discovered**. Results of comparing the approach of expanding all child nodes versus expanding only one child node, fewer states visited to discover the same number of modes, indicate higher exploration efficiency.

| States visited | 4 | 8 | 16 |
|---|---|---|---|
| MG2FlowNet (*all*) | **4,816** | **9,616** | **20,816** |
| MG2FlowNet (*one*) | 9,616 | 134,416 | / |

These experimental results conclusively validate our design rationale: expanding all valid child nodes during the expansion phase is essential for achieving optimal state visitation and exploration efficiency.

## G DETAILED EXPERIMENTAL SETUP

**Parameter Setup in Hypergrid Task.** For the GFlowNets policy model, we use the same configuration as vanilla GFlowNets, and we sampled trajectories with a batch size of 16, using the Adam optimizer with all other parameters at their default values. All experiments in this task are performed on a CPU. The horizon and dimension are set to 8 and 4. For the MCTS framework, we set the number of MCTS iterations to 1, the maximum depth of simulation to 20, the exploration coefficient to 1, and the greediness factor to 0.2.

**Parameter Setup in Molecule Design Task.** For the GFlowNets policy model, we use the dataset and proxy model provided by Bengio et al. (2021); Lau et al. (2024); Malkin et al. (2022). Different from the hypergrid experiment, due to the large state space of this experiment, we set the number of

MCTS iterations to 1, the maximum depth of simulation to 8, the exploration coefficient to 1, and the greediness factor to 0.2.

## H   PROOF OF EQUATION (10)

■ **Recall Equation (10):**

$$p_i = \frac{(Q_i - Q_{min})}{\sum_k (Q_k - Q_{min})}.$$

As shown in Eq. (10), instead of adopting a max-$Q$ strategy, we choose to use a softmax policy based on $Q$-values. This decision stems from our goal of encouraging more goal-directed behavior on top of a learned flow model, rather than pursuing greediness for its own sake. Directly using the maximal $Q$-value may lead to premature convergence and get stuck in a local optimum. Moreover, $Q$-values are often inaccurate in the early training stages, making the model highly sensitive to estimation errors. To mitigate these risks, we opt for a soft value policy $Q$ that more effectively balances exploitation and exploration.

■ **Proof.**   As for node $n$, we define the $Q$-values of the child nodes of node $n$ as a set $\{Q_1, Q_2, ..., Q_n\}$. In order to compute the $Q$-values distribution of these child nodes, we normalize the data of $Q$-values,

$$\hat{Q}_i = \frac{Q_i - Q_{min}}{Q_{max} - Q_{min}}. \tag{18}$$

Then, we need to obtain the probabilities of these child nodes. $P_i = \hat{Q}_i / \sum_k \hat{Q}_k$, so we can obtain this result:

$$p_i = \frac{(Q_i - Q_{min})/(Q_{max} - Q_{min})}{\sum_k (Q_k - Q_{min}/Q_{max} - Q_{min})}. \tag{19}$$

By normalizing both numerator and denominator through division by $Q_{max} - Q_{min}$, the formula can be simplified to:

$$p_i = \frac{Q_i - Q_{min}}{\sum_k (Q_k - Q_{min})}. \tag{20}$$

Therefore, we finally obtain Eq. (10).

## I   DISCUSSION

### I.1   LIMITATIONS

The present study is conducted in controlled environments where both the action space and the reward function remain fixed. This setting is sufficient for validating the core ideas of `MG2FlowNet` and provides a clear basis for comparison across methods. However, it does not cover scenarios where the available actions evolve during training or where the reward distribution changes over time. These cases are outside the scope of this work and will be investigated in future studies.

### I.2   FUTURE WORK

Building on the strengths of `MG2FlowNet`, a natural extension is to adapt the framework to more dynamic and realistic environments. One promising direction is to incorporate mechanisms that can flexibly accommodate evolving action sets, enabling the model to remain effective as the space of available choices expands or shifts. Another direction is to develop adaptive strategies for nonstationary reward distributions, where feedback signals change due to external interventions or shifting objectives. Possible solutions include adaptive exploration policies, meta-learning techniques that transfer knowledge across tasks, or hybrid methods that couple MCTS with fast bandit style estimators. Given the scalability and planning capabilities of `MG2FlowNet`, we believe these extensions would not only broaden its applicability but also strengthen its role as a general framework for lifelong reinforcement learning and adaptive molecular design.

## J USE OF LLMs

In preparing this manuscript, we used a large language model (LLM) as a writing assistant tool, specifically for grammatical refinement, style polishing, and correction of minor typographical errors. The LLM did not contribute to the scientific ideas, algorithm design, or experimental setup. All substantive content, reasoning, and conclusions are entirely the product of the authors. We accept full responsibility for all content in the paper, including parts refined or corrected by the LLM, and affirm that no text generated by the LLM constitutes original scientific contributions attributed to it.

