# OpenReview forum: "MG2FlowNet: Accelerating High-Reward Sample Generation via Enhanced MCTS and Greediness Control"
_ICLR.cc/2026/Conference — ICLR 2026 Conference Withdrawn Submission_

### Official Review · Reviewer_QmY3 · 2025-10-29

**Soundness:** 3
**Presentation:** 3
**Contribution:** 2
**Rating:** 2
**Confidence:** 4

**Summary:**

The paper proposes MG2FlowNet, which integrates MCTS into GFlowNets sampling to effectively balance exploration and high-reward samples discovery. The paper systematically analyzes how to integrate MCTS into GFlowNets and verifies its effectiveness in HyperGrid and Molecule discovery tasks.

**Strengths:**

- Over-exploration is one of the prevalent challenges when utilizing GFlowNets in real-world settings [1]. The paper tries to tackle this crucial challenge with MCTS, a powerful tool for balancing exploration and exploitation.

- Control of greediness is also a crucial point when designing a sampling policy for GFlowNets [2]. The paper tries to have a flexibility of control by estimating Q values.

[1] Kim, Minsu, et al. "Local Search GFlowNets." The Twelfth International Conference on Learning Representations.

[2] Lau, Elaine, et al. "Qgfn: Controllable greediness with action values." Advances in Neural Information Processing Systems 37

**Weaknesses:**

- The benchmarks are out of date. The paper conducts experiments on HyperGrid with one configuration (d=4, H=8) and the molecule design task. To verify the effectiveness of the method, it would be better to extend the experiments to more challenging scenarios like the deceptive grid suggested by [1], longer biological sequence design suggested by [2], and larger state space in combinatorial optimization tasks suggested in [3].

- For the baselines, several methods have been suggested to mitigate over-exploration, such as LS-GFN [2], PBP-GFN [4]. I strongly recommend comparing with those baselines or at least adding discussion sections for those works. While presenting the originality of the work is important, I believe that the discussion on several related papers is also crucial.

[1] Kim, Minsu, et al. "Adaptive teachers for amortized samplers." The Thirteenth International Conference on Learning Representations.

[2] Kim, Minsu, et al. "Local Search GFlowNets." The Twelfth International Conference on Learning Representations.

[3] Zhang, Dinghuai, et al. "Let the flows tell: Solving graph combinatorial problems with gflownets." Advances in Neural Information Processing Systems 36

[4] Jang, Hyosoon, et al. "Pessimistic backward policy for GFlowNets." Advances in Neural Information Processing Systems 37

**Questions:**

- While over-exploration is a major challenge in training GFlowNets, some argue that naive on-policy training of GFlowNets suffers from severe mode collapse, and we need to promote exploration. To this end, several papers have been suggested to promote exploration and achieve high mode coverage compared to the original GFlowNets [1-3]. Compared to those lines of work, under what conditions do we prefer the proposed method to exploration-promoting methods? I want to hear your opinion.

[1] Kim, Minsu, et al. "Adaptive teachers for amortized samplers." The Thirteenth International Conference on Learning Representations.

[2] Madan, Kanika, et al. "Towards improving exploration through sibling augmented gflownets." The Thirteenth International Conference on Learning Representations.

[3] Malek, Idriss, Abhijit Sharma, and Salem Lahlou. "Loss-guided auxiliary agents for overcoming mode collapse in gflownets." arXiv preprint

---

### Official Review · Reviewer_JzGM · 2025-10-30

**Soundness:** 2
**Presentation:** 1
**Contribution:** 1
**Rating:** 2
**Confidence:** 5

**Summary:**

This paper proposes MG2FlowNet, which uses MCTS with control of greediness in GFlowNets. It uses PUCT-based selection policy to tune greediness to balance exploration and exploitation. They experimented in two different tasks: Hypergrid and Molecule design. MG2Flownet outperforms baseline methods.

**Strengths:**

It tries to solve a core problem of GFlowNets, which suffer from sampling from multiple high-reward regions. It proposes a method that integrates some of the previously suggested methods that are proven to work well.

**Weaknesses:**

- Overall, the presentation of the paper is very weak. It is hard to follow the methods as it is. It seems like it needs a lot of revision to be accepted to ICLR.
- Neither using MCTS in GFN nor using Q-function in GFN is novel. This work tries to integrate the existing methods to create MG2FlowNet. However, it does not highlight the key difference or novel contribution compared to previous works ([1], [2])
- I strongly believe that this paper should have a preliminary section for details about MCTS or PUCT. I think the current Section 2 is just a repeat of the first few paragraphs of the introduction, so it would have been nicer if the authors had given some details about MCTS and PUCT here.
- Section 3 (Problem Formulation) is not rigorous at all. Not a single dimension is given for every variable or function used in the section.
- From lines 145 to 157, it is really hard to follow the framework. The authors mentioned "As illustrated in Figure 2," but none of the notations used in the framework paragraph appear in the figure. What is $s_0$ in the figure? What is $n_0$ or $G_m$? Is $G_m$ different from $\mathcal{G}_M$ from Section 3?

**Questions:**

- I think the motivation of this research is weak. You just mentioned two previous works ([1], [2]) and concluded that training GFN for high-reward regions is difficult. I believe there are a lot of works that have done this, including a simple trick of training temperature-conditional GFN ([3]). Can you please provide more comparisons with other methods while providing more extensive related works?
- In line 135, what do you expect readers to understand about "Our MCTS algorithm"? Did you ever mention it before or provide the algorithm right after? It does not make sense to assume the readers will understand this part as it is.
- In Section 4.1, why did you write $P_F$ in Equation 5? Should we understand it as $P_F(s|s')$ as defined in Equation 1? Can you please rigorously write the equation so that the variables used in the right-hand side appear as inputs on the left-hand side?
- In experiments for molecule design, how did you construct the reward? Did the authors mention what rewards are in the experiment sections? I think engineering the reward is very important for the performance of models.
- In experiments, what is the length of the sequence you are generating? I feel like the metric is quite arbitrary.
- How did the authors calculate the number of modes? If the authors just counted the number of generated samples that have a higher reward than a threshold, I think it is totally wrong. In order for samples to be "different modes", not only does it have high rewards, but the samples should be significantly different in terms of other metrics (either in sequence or structure space).
- Why do you think your method performs the worst when trajectories are shorter than ~3000? It seems like a critical drawback.
- I feel like the baseline methods are too sufficient nor powerful. Did you consider running other methods to compare with?

[1] Morozov, Nikita, et al. "Improving gflownets with monte carlo tree search." arXiv preprint arXiv:2406.13655 (2024).
[2] Lau, Elaine, et al. "Qgfn: Controllable greediness with action values." Advances in neural information processing systems 37 (2024): 81645-81676.
[3] Kim, Minsu, et al. "Learning to scale logits for temperature-conditional gflownets." arXiv preprint arXiv:2310.02823 (2023).

---

### Official Review · Reviewer_7RgS · 2025-10-31

**Soundness:** 2
**Presentation:** 2
**Contribution:** 2
**Rating:** 2
**Confidence:** 3

**Summary:**

This paper proposes MG2FlowNet, which integrates an MCTS-based planning module into Generative Flow Networks (GFlowNets) to improve the efficiency of discovering high-reward samples while maintaining diversity. The method combines PUCT-guided exploration with a tunable α-greedy mechanism that balances the contribution of Q-values from MCTS and the GFlowNet forward policy. Experiments on HyperGrid and molecule generation tasks demonstrate faster discovery of high-reward modes compared to existing baselines such as TB, SubTB, and QGFN.

**Strengths:**

Strengths

1. Clear motivation — addresses the long-standing issue that vanilla GFlowNets tend to overexplore low-reward regions during early training.

2. Methodical clarity — the paper is well organized, and the integration of PUCT and α-greedy sampling is easy to follow.

3. Empirical improvement — the method demonstrates tangible gains on two representative benchmarks (HyperGrid and molecule design), particularly in early high-reward discovery speed.

**Weaknesses:**

Weaknesses

1. Incremental novelty

- The proposed approach appears to be a modest extension of QGFN, where the Q-value is already used to control greediness during off-policy exploration.

- MG2FlowNet mainly replaces the learned Q-value estimator with MCTS-derived Q-values, without introducing fundamentally new theoretical insights.

2. The idea of using a search-based planner to bias exploration is also reminiscent of Local Search GFlowNet, Genetic-Guided GFlowNet, and other off-policy exploration variants, none of which are compared experimentally or even discussed in detail. As a result, the methodological contribution feels like an incremental engineering improvement rather than a new conceptual advance.

3. Weak experimental validation

- The evaluation focuses only on reward-based metrics such as top-k average reward and Tanimoto similarity.

- However, probabilistic metrics fundamental to GFlowNets (e.g., ELBO, EUBO, KL divergence to the target reward-proportional distribution) are missing.

- Without these, it is unclear whether MG2FlowNet actually improves the quality of the learned distribution or merely biases it toward a few high-reward modes.

- Additionally, the molecule design benchmark is too limited in scale and diversity to support strong claims.

3. Scalability and stability concerns

- The paper provides no analysis of computational cost or scalability of MCTS when integrated into large-action GFlowNets.

- MCTS rollouts scale poorly with branching factor, and the paper does not analyze runtime or memory impact.

- There is also no evidence that mode collapse is avoided under large-scale or high-dimensional settings.

- The $\alpha$ parameter introduces an additional degree of freedom, yet there is no principled or empirical guidance on how to tune it across tasks; the ablation in Table 2 is minimal and insufficient.

4. Incomplete related work discussion

- The GFlowNet literature is broad, covering off-policy, amortized, variational, and evolutionary approaches (e.g., Local Search GFN, Evolutaionary GFN, MetaGFN, Genetic-guided GFN).

- The connection to recent advances in variational inference formulations is also underdeveloped.

**Questions:**

1. How does MG2FlowNet perform under large action spaces (e.g., ≥10⁵ actions per step) where MCTS becomes computationally expensive?

2. Could the authors report ELBO or EUBO values to confirm whether the learned distribution still approximates the target reward-proportional distribution?

3. How sensitive is performance to the choice of α and cₚᵤcₜ? Is there any heuristic or adaptive schedule that generalizes across tasks?

4. How does MG2FlowNet compare to Local Search GFlowNet or Genetic-Guided GFlowNet in terms of both sample efficiency and mode coverage?

5. Is the Q-value in MCTS updated with bootstrapped returns or full rollout rewards? How does that interact with the flow balance constraint?

---

### Official Review · Reviewer_T84V · 2025-10-31

**Soundness:** 3
**Presentation:** 3
**Contribution:** 3
**Rating:** 4
**Confidence:** 4

**Summary:**

MG2FlowNet, a GFlowNet sampler that integrates PUCT-guided MCTS with a tunable α-greedy mixing of MCTS action values and the GFlowNet forward policy PF. Basically:

- selection uses a PUCT rule to balance exploration and exploitation;
- expansion adds all legal children;
- simulation follows PF;
- backprop only updates nodes along the selected path;
- action choice mixes PF with a normalized Q-distribution controlled by α.

**Strengths:**

Simple but effective greediness control; empirical gains in important area (on molecules);

**Weaknesses:**

- Non-proportional sampling objective drift. The method explicitly accepts larger L1 error (deviating from reward-proportional sampling) in exchange for high-reward focus. This is a philosophical departure from a core GFlowNet goal and should be framed as a different objective more directly.
- MCTS overhead is discussed qualitatively (e.g., expand-all children), but there’s no wall-clock/throughput breakdown or memory profile vs. baselines, especially for large vocabularies in molecules.
- Ablation: critical knobs (MCTS rollout depth/iterations, simulation policy variants, mixing schedules) aren’t systematically explored beyond a brief temperature schedule that underperforms.
- There’s no theory connecting the modified sampler to reward-proportionality or bounding the induced bias/diversity loss; the work remains empirical.

**Questions:**

- If the end goal is not exact reward-proportional sampling, what is the formal target distribution? Can the authors define and measure “high-reward-biased yet diverse” distributions beyond top-k metrics?

- PUCT depends on accurate visit counts and priors; with one MCTS iteration in experiments, how much guidance is MCTS really providing vs. just re-weighting PF? Please quantify performance vs. number of MCTS playouts and the marginal cost/benefit.

- Diversity beyond fingerprints: Tanimoto similarity over top-1000 molecules is helpful, but can the authors report scaffold diversity and novelty vs. training data, and examine any correlation between α and mode collapse risk?

- (Not required, Bonus) Apple-to-apple comparison: QGFN relies on Q-values that can be noisy early; MG2FlowNet claims better early behavior via the exploration term. Could QGFN + calibrated exploration (e.g., optimistic init or uncertainty-aware Q) close the gap? A controlled study would strengthen the claim.

---

### Note · Authors · 2026-01-24

I have read and agree with the venue's withdrawal policy on behalf of myself and my co-authors.